# HPLC-Based Analysis of Impurities in Sapropterin Branded and Generic Tablets

**DOI:** 10.3390/pharmaceutics12040323

**Published:** 2020-04-03

**Authors:** Emanuela Scudellaro, Luciana Tartaglione, Fabio Varriale, Carmela Dell’Aversano, Orazio Taglialatela-Scafati

**Affiliations:** Department of Pharmacy, School of Medicine and Surgery, Universita’ degli Studi di Napoli Federico II, Via D. Montesano, 49, 80131 Naples, Italy; scudomanu@gmail.com (E.S.); luciana.tartaglione@unina.it (L.T.); fabio.varriale@unina.it (F.V.); dellaver@unina.it (C.D.)

**Keywords:** sapropterin, PKU, BH_4_ deficiency, chemical content, impurity identification, HPLC-UV, MS/MS

## Abstract

This work was aimed at the definition of a chromatographic method able to separate and quantify impurities present in sapropterin-containing drugs during an accelerated stability study. The chromatographic method was applied to the orphan drug Kuvan^®^ and to its corresponding generic sapropterin Dipharma (Diterin^®^), both of which are approved for the treatment of hyperphenylalaninemia-induced symptoms. The two products tested had a similar manufacture date and both had an approved stability shelf-life of three years. Samples were analyzed by HPLC at *T* = 0 and after six months of storage at 40 °C and 75% relative humidity. Identification of the impurities was supported by a detailed mass spectrometry and MS/MS profile. The analysis demonstrated an overall higher stability for the Diterin^®^ formulation, which was related to a lower increase of some impurities compared to Kuvan^®^.

## 1. Introduction

Sapropterin dihydrochloride, (6*R*)-L-erythro-5,6,7,8-tetrahydrobiopterin dihydrochloride (BH_4_·2HCl), referred also as sapropterin, is a synthetic version of the naturally occurring 6*R*-BH_4_, which is, among others, a cofactor of phenylalanine-, tyrosine-, and tryptophan hydroxylases. Sapropterin is the active ingredient of the drug Kuvan^®^, a medicine approved by the Food and Drug Administration (FDA) in 2007 and the European Medicines Agency (EMA) in 2008, for the treatment of hyperphenylalaninaemia (HPA), a rare disease caused by defects in the phenylalanine hydroxylase (PAH) [1] or by a defect in the biosynthesis or recycling of BH_4_. Mutations in the phenylalanine hydroxylase gene (PAH) lead to phenylketonuria (PKU) which, depending on the seriousness of the mutations, can be mild (Phe levels 300–600 µM), moderate (Phe levels 600–1200 µM), or severe (Phe levels >1200 µM). On another side, defects in BH_4_ synthesis or recycling lead to BH_4_ deficiency. The latter occurs in only around 2% of the HPA patient population [2]. In Europe, patients who are found to respond to a seven-day BH_4_ loading test (30% decrease of phenylalanine level in the bloodstream), could benefit from a life-long treatment with Kuvan^®^, at a daily dose of between 5 and 20 mg/kg [3].



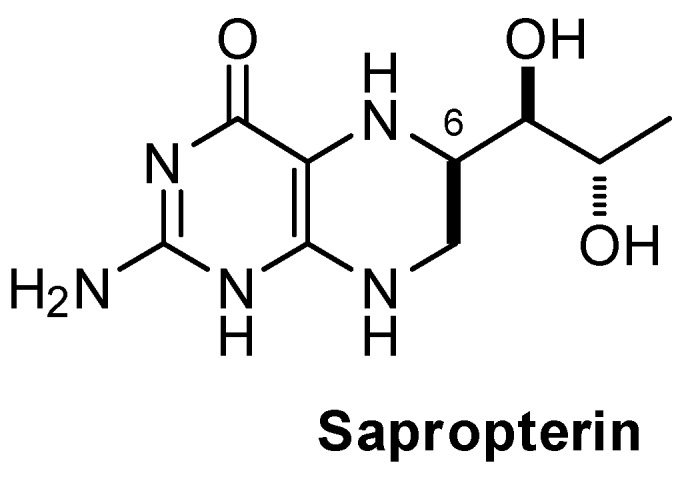



Sapropterin is a relatively complex and unstable molecule, whose synthesis is made difficult by the presence of three adjacent asymmetric centers. Until 1999, sapropterin was obtained as a mixture of 6*R*/6*S* diastereomers in a 69:31 ratio. However, only the 6*R* isomer is biologically active, the 6*S* isomer has been reported to cause an irreversible inactivation of rat liver PAH [4]. HPA commonly requires life-long treatment; therefore, the medicine must be of the highest quality and stability in order to avoid the ingestion and accumulation of potentially dangerous impurities over the patient’s life. Over the years, some papers have raised doubts about the quality of some generic drugs, which are indicated to be of lower quality compared to their branded counterparts [5]. Such papers prompted FDA Commissioner, Dr Scott Gottlieb, and the Center for Drug Evaluation and Research (CDER) Director, Dr Janet Woodcock, to publish a statement to deny any alleged lower safety and/or effectiveness of FDA approved generics versus their corresponding branded drugs [6]. Notwithstanding this initiative, the feeling that branded drugs cannot indiscriminately be exchanged by their corresponding generics, is widely spread among practitioners and some patient communities.

The aim of the present study was to experimentally verify this perception by investigating the composition and stability of the life-long treatment sapropterin-containing drug Kuvan^®^ and its generic sapropterin Dipharma (Diterin^®^). Furthermore, our data, based on HPLC-UV and LC-high resolution mass spectrometry (LC-HRMS) analysis, moved a step forward in providing the most detailed profile available to date of the impurities present in sapropterin-containing products, leading to the identification of nine by-products.

## 2. Materials and Methods

### 2.1. Material

Kuvan^®^, batch number L141191, was purchased in an ordinary manner from a pharmacy with expiry date: 02/2020. The manufacture date was not available; however, the leaflet of Kuvan^®^ indicated a 3-year shelf-life indicating a manufacture date of around February 2017. Diterin^®^ was generously donated by Dipharma SA (Chiasso, Switzerland), batch No. 17107H2, manufacture date: 05/2017, and expiry date: 04/2020. Starting from *T* = 0, both products were stored at 40 °C and 75% Relative Humidity (RH) in a constant climate chamber model HPP110 (Memmert, Italy), *V* = 108 L. Samples of both products were -taken at *T* = 0 and 6 months for the purpose of analysis. Storage at 40 °C and 75% RH for 6 months are the conditions indicated by EMA for accelerated stability studies (Note for Guidance on Stability Testing, CPMP/ICH/2736/99).

Water HPLC Plus, acetonitrile for HPLC (≥99.9%), ammonium formate (reagent grade, 97%), formic acid (reagent grade, ≥95%), and sapropterin, were all purchased from Sigma Aldrich (Milan, Italy). Sodium dihydrogen phosphate dihydrate (NaH_2_PO_4_·2H_2_O), phosphoric acid 85% (H_3_PO_4_), and ascorbic acid were purchased from Titolchimica S.p.A (Pontecchio Polesine, Italy). HPLC-UV analysis was carried out on a Shimadzu LC-10AD instrument equipped with detector, Shimadzu SPD-10A (Shimadzu Italia, Milan, Italy). A clarity software program was used for the chromatogram analysis.

### 2.2. Methods

#### 2.2.1. HPLC

Kuvan^®^ and Diterin^®^ tablets were pulverized, after which 7.0 mg of the obtained fine powders were accurately weighed, transferred into test tubes, and dissolved in 7.0 mL of water containing 0.2% ascorbic acid (*w/v*). Given the reported solubility of sapropterin in water (2.0 mg/mL), we assumed that it was fully solubilized in these conditions; however, in order to eliminate insoluble excipients, the solution was sonicated in an ice bath for 15 min with intermittent shaking, and then centrifuged at 3000 rpm (1.21× *g*) for 20 min (MPW-56 centrifuge, MPW instruments, Warszawa, Poland). The clear supernatant was collected and 20 µL was immediately (roughly 40 min after tablet pulverization) analyzed by HPLC-UV. Chromatographic separation was performed in the isocratic mode, using an ion-exchange Partisil^®^ column 10 SCX-250A, 250 × 4.6 mm i.d, 5 µm particle size. Mobile phase was 0.03 M NaH_2_PO_4_ solution—pH adjusted to 3.0 with phosphoric acid (H_3_PO_4_). The flow rate was set at 1.2 mL/min and the run time was 20 min. The injection volume was 20 µL and the UV detector was set at 265 nm. The percentage concentration of the impurities was calculated as follows:% Concentration = (A_p_ / A_t_) × 100
where *A*_p_ = area of the impurity peak, *A*_t_ = total area of the impurities and sapropterin.

Three independent measurements were carried out for each sample.

#### 2.2.2. Mass Spectrometry

LC-HRMS experiments were carried out on a Dionex Ultimate 3000 system, which included a solvent reservoir, in-line degasser, a quaternary pump and refrigerated autosampler, and column oven coupled to a hybrid linear ion trap LTQ Orbitrap XLTM Fourier Transform MS (FTMS) equipped with an Electrospray (ESI) ION MAXTM source (Thermo-Fisher, San Josè, CA, USA). Separations were performed on a 150 mm × 2.0 mm ID column packed with 3 µm TSK-GEL^®^ Amide-80 material (Sigma Aldrich) at room temperature. Mobile phases were (A) water and (B) 95% acetonitrile and 5% water (*v/v*), both containing 2.0 mM ammonium formate and 3.6 mM formic acid (pH 3.55). A gradient elution, setting both flow rate and injection volume at 0.2 mL/min and 5 µL respectively, was run starting from 90% B (*t* = 0) and held for 20 min (*t* = 20), it was then decreased from 90% to 20% B in 1 minute (*t* = 21) and held for 10 min (*t* = 31), it was then increased to 90% B in 1 minute (*t* = 32) and held for 10 min (*t* = 42) to re-equilibrate the column. A standard solution of 6*R*-(BH_4_) at 10 µg/mL (ascorbic acid 0.2% *w/v*) was prepared, and used to set up the source parameters while operating in a positive ionization mode, as follows: capillary temperature = 360 °C, sheath = 46 and auxiliary gas = 7.5 (arbitrary units) spry voltage = 4.8 kV, capillary voltage = 24 V, and tube lens = 70 V. Full scan spectra were recorded in the range *m/z* 150–300. HR collision-induced dissociation (CID) experiments were carried out selecting the [M+H]^+^ ion of each compound as precursor and collision energies (CEs) (depending on the precursor ion), in the range of 22% to 25%, isolation width = 2.0, activation *Q* = 0.250, and activation time = 30 ms. In all the experiments, the resolving power (RP) was set at 60,000 (FWHM at *m/z* 400). Elemental formulae were calculated using the mono-isotopic ion peak of each cluster through Thermo Xcalibur software v2.2 SP1.48 (Thermo Fisher, San Josè, CA, USA) at a 5 ppm mass tolerance. A full scan HRMS spectrum of BH_4_ standard solution, showed the characteristic and dominant [M+H]^+^ ion at *m/z* 242.1248 (C_9_H_16_N_5_O_3_, Ring Double Bond Equivalent RDB = 4.5, Δppm = 0.637), and the relevant [M+Na]^+^ adduct ion at *m/z* 264.1069 (C_9_H_15_N_5_O_3_, RDB = 4.5, Δppm = 0.717) with a relative abundance of 80–90% for the pseudomolecular ion. A mixture containing each standard at 10 µg/mL (in water containing ascorbic acid 0.2% *w/v*) was prepared and injected under the same experimental conditions. Then, the LC-HRMS spectrum was acquired, and through its interpretation, as reported for BH_4_, the presence of the peculiar [M+Na]^+^ ion was highlighted. However, its relative abundance, with respect to the relevant [M+H]^+^ ion, was characteristic for each compound.

## 3. Results

### 3.1. Individual Impurity Identification

Finely pulverized, Kuvan^®^ and/or Diterin^®^ tablets were dissolved in 7.0 mL of a 0.2% (*w/v*) solution of ascorbic acid in water, sonicated, centrifuged and analyzed by HPLC-UV in isocratic mode on an ion-exchange Partisil^®^ column 10 SCX-250A, 250 × 4.6 mm i.d, 5 µm, using 0.03 M NaH_2_PO_4_ water solution, with pH adjusted to 3.0. Peaks were detected with the UV detector set at λ 265 nm. This analytical method was selected on the basis of its capability to separate and detect nine impurities/degradation products, providing better results compared to other methods present in the literature [7,8]. Figure 1 reports the HPLC profile obtained for Kuvan^®^ tablet at *T* = 0. The two major peaks were easily identified as ascorbic acid (RT = 2.94 min) and sapropterin (RT = 8.61 min), and confirmed by injection of standards.

Nine minor compounds (**1**–**9**) attributable to impurities/degradation products were detected at retention times ranging from RT = 3.68 to 15.1 min. The origin of these compounds could be attributable either to small impurities in the sapropterin, deriving from its synthetic procedure, or to products of its degradation. During the last few decades, sapropterin has been the subject of intense investigation aimed at identifying its degradation products, which mainly originate from oxidation and hydrolysis [7,8], and these products are the obvious candidates for the assignment of the structures of compounds **1**–**9**. Since the majority of them were commercially available, injection of the standards in the same HPLC conditions, allowed for the confident identification of the compounds as reported in Table 1. In particular, compound **6** was assigned as (6*R*)-tetrahydrobiolumazine, since it co-eluted with the commercially available product of sapropterin hydrolysis, while the isomeric compound **7**, was assigned as (6*S*)-tetrahydrobiolumazine by comparison with the synthetic biolumazine mixture, which was obtained by following procedures from the literature [9,10]. Similarly, compound **3** was assigned as 7,8-dihydrobiopterin by comparison with the commercially available standard, while the isomeric compound **4** was tentatively assigned as 5,6-dihydrobiopterin. The assignment of compounds **1**–**9** was supported by a detailed LC-HRMS and (MS/MS) analysis carried out on standards and on the tablet (see Section 3.2).

### 3.2. Mass Spectrometry Analysis

Although a few liquid chromatography-tandem mass spectrometry methods (LC-MS/MS) are available to determine tetrahydrobiopterin [11,12,13], there is shortage of spectrometric data acquired on high resolution MS systems. Therefore, LC-HRMSn (*n* = 1,2) spectra of each available impurity (**1**–**3** and **5**–**9**) were acquired and analyzed in order to investigate the fragmentation pattern. Mobile phase selection was made taking into account the pH used for the above described LC-UV detection (pH = 3.5). Considering that the use of a phosphate buffer is not compatible with ESI-MS ionization and that the analyzed compounds are small-size highly polar molecules, hydrophilic interaction liquid chromatography (HILIC) was selected to chromatographically resolve these compounds. Each standard was separately injected onto the column at a concentration level of 10 µg/mL and the full HRMS spectra obtained, were in all cases dominated by the [M+H]^+^ ion with the presence of a minor sodium adduct (Appendix A). Each standard was analyzed within 90 min from sample preparation in order to minimize degradation phenomenon that would result in the formation of oxidation products and a consequent dramatic signal decrease. Figure 2 reports Extracted Ion Chromatograms (XICs) of the standards, obtained by selecting the relevant exact masses of each compound. Collision induced dissociation (CID) HR-MS2 spectra associated with each peak were also acquired (Appendix A), and the elemental compositions for all the obtained fragment ions are reported in Appendix A. Kuvan^®^ and Diterin^®^ tablets were then dissolved using the procedure reported in the Materials and Methods Section 2.2.2 and analyzed versus the standards under the same experimental conditions.

### 3.3. Comparative Analysis of Kuvan^®^ and Diterin^®^ Tablets

Having identified the nature of nine impurities present in sapropterin-containing tablets, we next employed the above developed HPLC method to the comparative analysis of Kuvan^®^ and Diterin^®^ tablets. Each product was analyzed at two different time points (*T* = 0 and 6 months) to quantify each single impurity, evaluate the total impurity amount and its trend over the timeframe. Representative examples of the chromatograms obtained at different times for different products are reported in the Appendix A.

Table 2 reports the percentage concentration of the impurities for each sample, calculated as (*A*_p_/*A*_t_) × 100, where *A*_p_ = area of the impurity peak and *A*_t_ = total area of the impurities and sapropterin. Data are presented as the mean ± SD of at least three independent measurements carried out for each sample. Peak areas have been normalized to sapropterin absorption on the basis of the UV response at 265 nm for each compound (the following ε at 265 nm in 10^3^ × M^−1^·cm^−1^ have been estimated 1 = 14.8; 2 = 12.3; 3 = 10.5; 4 = 10.2; 5 = 13.0; 6 and 7 = 10.4; 8 and sapropterin = 11.0; 9 = 10.8).

## 4. Discussion

Sapropterin is an effective agent in lowering blood hyperphenylalaninaemia (HPA) in patients with phenylketonuria (PKU) or BH_4_ deficiency, and, to date, no other natural or synthetic agent has demonstrated the same efficacy. Sapropterin is formulated for oral administration in tablets (or powder in sachets) that are dissolved in water prior to administration. The active pharmaceutical ingredient (API) is obtained synthetically through a somewhat complex route, with the main difficulty residing in controlling the configuration of three asymmetric centers. The processes giving rise to the API present in Kuvan^®^ and Diterin^®^, enable the obtention of the 6R form with high purity, although very small amounts of the undesired 6S stereoisomer may be present. It is to be noted that the 6S form is not only inactive, but it may cause inactivation of phenylalanine hydroxylase.

Sapropterin is a relatively unstable compound and its degradation chemistry is very complex, mostly ascribable to the sensitivity of side chain and ring moiety to oxidation. Tautomeric equilibria on the oxidized products render even more complex a comprehensive understanding of the exact formation and quantification of all the impurities. It has been estimated that in a sample of sapropterin left in the open, traces of biopterins **3** and **4** will start to appear after only 15 min incubation, and after 30 min, the amount of sapropterin is halved [11]. To minimize the impact of these reactions, sapropterin-based tablets contain the antioxidant ascorbic acid.

As with any other medicine, sapropterin tablets can contain minor impurities deriving from their synthetic procedure as well as products formed upon degradation of the API itself. The aims of this work were: (a) to identify a separation method able to efficiently separate and quantify as many impurities as possible; (b) to apply this method to evaluate the impurities of commercially available sapropterin-containing tablets; and (c) to compare the impurity profile of generic and branded sapropterin-containing tablets, and finally evaluate them over time. To our knowledge, the method reported above, based on the use of an ion-exchange HPLC column, is one of the most efficient reported to date, as it enables the efficient separation and detection of nine impurities/degradation products from sapropterin-containing tablets.

Data reported in Table 2, evidence that the tablets contain an impurity percentage concentration ranging from 0.3% to 0.7%. If we had also taken into account the stabilizer ascorbic acid and the other excipients (mainly mannitol), the impurity percentage concentration would have been obviously further lowered. Figure 3 schematically depicts the trend of impurity concentration for the examined products, when stored at 40 °C and 75% RH. As expected, an increase in the impurity level is observed over time for both tested commercial products, which can be clearly ascribed to an increase in the concentration of degradation products. The detailed trend of impurities is complex, and its full rationalization is extremely difficult; however, a more detailed analysis of the behavior of some selected impurities can better clarify this point.

The concentration of compound **1** is practically constant over the time and this could be explained by its origin as a synthetic impurity of the active principle sapropterin. The assignment of this peak fully supports this hypothesis since it has been assigned as biopterin, a key intermediate in the syntheses of sapropterin. The average level of this impurity is in the same range for both Kuvan^®^ and Diterin^®^.

Figure 4 reports a likely, although necessarily simplified, conversion scheme that could help to rationalize the formation of the other compounds. Biolumazines (**6** and **7**) are the products of hydrolysis of sapropterin (or of its epimer), resulting in the substitution of the amino group on the pyrimidine ring with a hydroxy group and subsequent tautomerism. Dihydrobiopterins **3** and **4** are regioisomeric oxidation products of sapropterin [14], and the trend of their concentrations is interesting. The concentration of **3** decreased over the time, probably because this compound was further oxidized to sepiapterin (**2**), thus concurring to the marked increase at *T* = 6 of **2**. In our experiments, at *T* = 6, sepiapterin **2** was the major impurity in both products, accounting for 0.169% in Diterin^®^ and 0.221% in Kuvan^®^. In Diterin^®^ sepiapterin concentration actually increased more than 4-fold, meanwhile 7,8-dihydrobiopterin (**3**) decreased 3-fold. It was interesting to note that the same trend, even if little bit more accentuated, was found in Kuvan^®^. On the other hand, direct oxidation of sapropterin to **4** caused an increase of this degradation product at *T* = 6 (2-fold more intense as in Kuvan^®^), although, in turn, the oxidation of compound **4** could provide a further source of pterin (**5**) and biopterin (**1**). Impurity **5** showed a marked increase at *T* = 6 in Kuvan^®^, while only a small increase was noticed in Diterin^®^. In Kuvan^®^, a parallel decrease of biopterin (**1**) at *T* = 6 was observed, therefore a conversion of **1** into **5** may be hypothesized in this sample.

A highly undesired impurity is (6*S*)-sapropterin epimer (**8**), which is capable of irreversibly inactivating the PAH enzyme [4]. Although both products contained a very little amount of this undesired side-product, after six months of storage in accelerated conditions, Diterin^®^ demonstrated a roughly 50% inferior amount of **8** with respect to Kuvan^®^. Finally, 5,6,7,8-tetrahydropterin (**9**) could derive from a direct cleavage of the side chain of sapropterin and this impurity is present at *T* = 6 in higher concentrations in Diterin^®^ than in Kuvan^®^. Grouping together all the impurities, we can observe that Kuvan^®^ ranges from 0.389% (*t* = 0) to 0.694% (*t* = 6) (78% increase), while Diterin^®^ ranges from 0.285% (*t* = 0) to 0.395% (*t* = 6) (39% increase).

## 5. Conclusions

In conclusion, data from an efficient HPLC-based analysis, able to separate and detect nine impurities/degradation products, showed an overall lower level of impurities/degradation products and higher stability of the sapropterin-containing generic tablets Diterin^®^ formulation, which mostly relate to a lower increase of impurities **2**, **4** and **6**, compared to Kuvan^®^. The lower increase of the efficacy-reducing impurity **8**, is also worthy of being mentioned, because although it is numerically less significant, it is potentially clinically significant.

We are well aware that a detailed comparison between the two commercial products would require further investigation; it is, nevertheless, undisputable that both Kuvan^®^ and Diterin^®^ completely conform to the standard quality of the major regulatory authorities, even after six months storage in accelerated stability conditions.

## Figures and Tables

**Figure 1 pharmaceutics-12-00323-f001:**
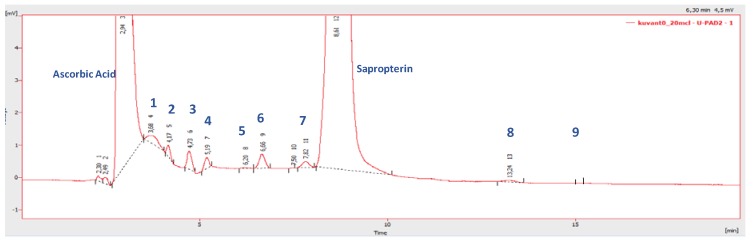
HPLC profile of pulverized Kuvan^®^ tablet at *T* = 0 dissolved in 7.0 mL of a 0.2% (w/v) solution of ascorbic acid in water. HPLC-UV (λ 265 nm) was run in isocratic mode on an ion-exchange Partisil^®^ column using 0.03 M NaH_2_PO_4_ water solution (pH 3.0) as an eluent.

**Figure 2 pharmaceutics-12-00323-f002:**
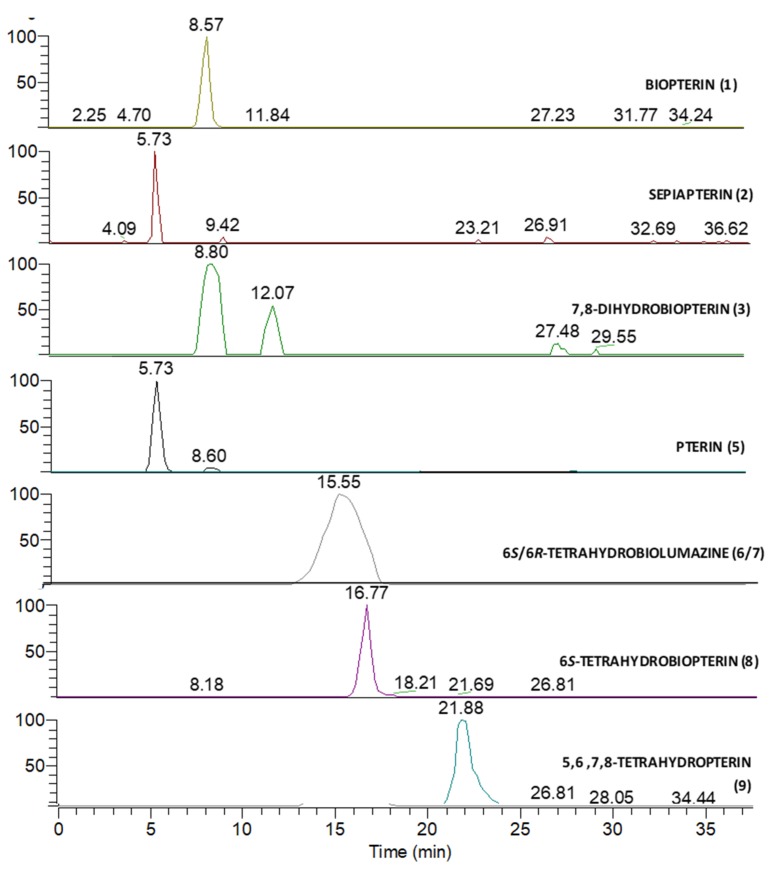
Extracted ion chromatogram (XIC) of compounds obtained by selecting exact masses of [M+H]^+^ and [M+Na]^+^ ions of each compound (see Appendix A).

**Figure 3 pharmaceutics-12-00323-f003:**
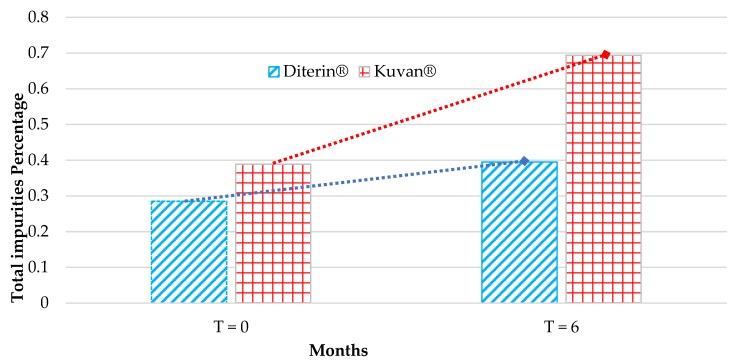
Trend of total impurities concentration (Table 2) for Kuvan^®^ and Diterin^®^ at *T* = 0 and after 6 months at 40 °C and 75% RH (accelerated stability study).

**Figure 4 pharmaceutics-12-00323-f004:**
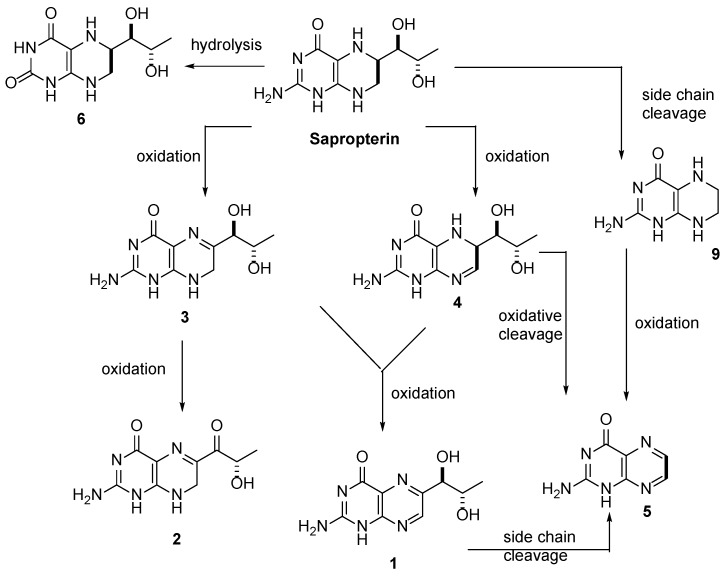
Proposed conversion scheme for the sapropterin degradation products detected in the HPLC-UV analysis.

**Table 1 pharmaceutics-12-00323-t001:** Assignment of minor compounds detected for Kuvan^®^ tablet at T = 0.

Peak Number	Name	Chemical Structure	Standard RT (min)	Sample RT (min)
**1**	Biopterin	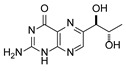	3.65	3.68
**2**	Sepiapterin	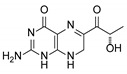	4.16	4.17
**3**	7,8-dihydrobiopterin	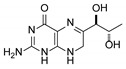	4.74	4.73
**4**	5,6-dihydrobiopterin	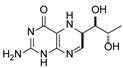	5.20	5.19
**5**	Pterin	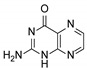	6.21	6.20
**6**	6*R*-tetrahydrobiolumazine	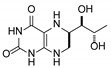	6.63	6.66
**7**	6*S*-tetrahydrobiolumazine	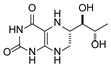	7.77	7.82
**8**	6*S*-tetrahydrobiopterin	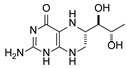	13.22	13.24
**9**	5,6,7,8-tetrahydropterin	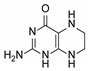	14.99	15.01

**Table 2 pharmaceutics-12-00323-t002:** Percentage concentration of the impurities for Kuvan^®^ and Diterin^®^ tablets at different times. Peak areas of ascorbic acid and sapropterin have been omitted. Data are presented as the mean ± SD of at least three independent measurements carried out for each sample.

Peak Number	Kuvan^®^ *T* = 0	Diterin^®^ *T* = 0	Kuvan^®^ *T* = 6	Diterin^®^ *T* = 6
**1**	0.058 ± 0.008	0.031 ± 0.002	0.027 ± 0.002	0.030 ± 0.003
**2**	0.035 ± 0.006	0.037 ± 0.003	0.221 ± 0.012	0.169 ± 0.007
**3**	0.091 ± 0.004	0.101 ± 0.008	0.037 ± 0.004	0.031 ± 0.004
**4**	0.051 ± 0.002	0.032 ± 0.003	0.176 ± 0.009	0.050 ± 0.005
**5**	0.004 ± 0.001	0.003 ± 0.000	0.019 ± 0.002	0.004 ± 0.004
**6**	0.090 ± 0.007	0.011 ± 0.001	0.135 ± 0.009	0.010 ± 0.001
**7**	0.042 ± 0.001	0.009 ± 0.001	0.045 ± 0.003	0.006 ± 0.001
**8**	0.018 ± 0.001	Not detectable	0.026 ± 0.003	0.015 ± 0.003
**9**	Not detectable	0.061 ± 0.004	0.008 ± 0.001	0.080 ± 0.006
**Total**	**0.389 ± 0.0046**	**0.285 ± 0.0039**	**0.694 ± 0.0066**	**0.395 ± 0.011**

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
