# Peer review of "HPLC-Based Analysis of Impurities in Sapropterin Branded and Generic Tablets"

_pharmaceutics, 2020, doi:10.3390/pharmaceutics12040323_

Round 1
Reviewer 1 Report
The authors report on the chromatographic method developed for quantification of impurities in sapropterin containing drugs (Kuvan and Dipharma) during an accelerated stability study. The authors have performed comprehensive chromatographic (HPLC-UV, and HPLC-HRMS) study for mapping impurity profiles which are present in the sapropterin-containing drugs, therefore providing a significant contribution to the scientific community. The authors have identified the nine compounds comprehensively, following the use of standard compounds as well as confirming the mass fragmentation pattern. The paper is well written, and the references are properly cited. Therefore, the paper can be published in the present form. There are only few minor comments.
Table 3. In the column total, standard deviations are wrongly calculated. Basically, they are highly overestimated. Instead of simple addition (standard deviations cannot be simply summed up, but variances can), they should be calculated as the squared root of the average of variances. In that way you should get the following values of standard deviations, arranged in the order of columns: 0.0046 (instead of 0.03), 0.0039 (instead of 0.022), 0.0066 (instead of 0.044), and 0.011 (instead of 0,034).
It is given in the text (line 196). However, if you do not mind, please provide the info regarding the result uncertainty in the table title as well. In this way it would be much easier for readers to follow.
Author Response
Reviewer 1
Thank you for your positive opinion on our work
Table 3. In the column total, standard deviations are wrongly calculated. Basically, they are highly overestimated. Instead of simple addition (standard deviations cannot be simply summed up, but variances can), they should be calculated as the squared root of the average of variances. In that way you should get the following values of standard deviations, arranged in the order of columns: 0.0046 (instead of 0.03), 0.0039 (instead of 0.022), 0.0066 (instead of 0.044), and 0.011 (instead of 0,034).
Answer: You are perfectly right. We have made this correction in Table 3 (now Table 2)
It is given in the text (line 196). However, if you do not mind, please provide the info regarding the result uncertainty in the table title as well. In this way it would be much easier for readers to follow.
Answer: Uncertainty statement of line 189-190 has been copied in the Table 3 (now Table 2) title
Reviewer 2 Report
The authors are presenting an analytical chromatographic method to separate and quantify impurities found in sapropterin-containing drugs (Diterin and Kuvan). HPLC and MS analysis are the tools used.
Authors could illustrate the effect of impurities on sapropterin-containing drugs` efficacy. The two drugs expired after 3 years. If the authors investigated the effect of storage for only 6 months, so what is the impact of impurities on drugs safety and efficacy after 3 years. HPLC profiles for sapropterin and sapropterin-containing drugs (Diterin and Kuvan ) at T= 0, and after 6 months of storage should be included into the manuscript. In Fig 3S: Some impurities disappeared with longer storage time, could you give more explanation. A comprehensive discussion should be added in order to interpret the findings of this work. Figures captions need more explanation. References should be updated In the analysis step, the author should mention or test the amount of the drugs after the sonication, and centrifuge processes. Changing from ascorbic acid-containing water to formic acid-containing water in MSMS should be discussed in more detail. In such work, the solution of the 6R-(BH4) used as a source parameter. Please also check if it has impurities or containing other isomer or not. The authors should provide the analysis data. The coming reference may help in the method and analysis part.
M Farag, D Rasheed; A Porzel, A Frolov, HR El-Seedi, LA Wessjohann (2018): Comparative analysis of Hibiscus sabdariffa (roselle) hot and cold extracts in respect to their potential for the inhibition of alpha-glucosidase. Food Chemistry, 250, 226-244.
Author Response
Reviewer 2
The two drugs expired after 3 years. If the authors investigated the effect of storage for only 6 months, so what is the impact of impurities on drugs safety and efficacy after 3 years.
Answer: This was an accelerated stability study. Storage in a climate chamber at 40°C and 75% Relative Humidity for 6 months are the conditions indicated by EMA for accelerated stability studies (Note for Guidance on Stability Testing, CPMP/ICH/2736/99). This point has been more clearly specified in lines 68-70.
HPLC profiles for sapropterin and sapropterin-containing drugs (Diterin and Kuvan ) at T= 0, and after 6 months of storage should be included into the manuscript.
Answer: These figures are available to the reader as Supporting Material. We would prefer to avoid including them into the main manuscript, that would become too long. In addition, the quantitative data about impurity concentrations are fully reported in Table 3 (now Table 2).
In Fig 3S: Some impurities disappeared with longer storage time, could you give more explanation. A comprehensive discussion should be added in order to interpret the findings of this work.
Answer: Table 3 (now Table 2) provides a detailed account of the percentage concentration for each impurity over the time. Inspecting this Table, only compounds 1 (for Kuvan) and 3 (for both Kuvan and Diterin) significantly decrease over the time. We have discussed the formation and interconversion of impurities in the Discussion paragraph from lines 245 to 273 and illustrated it in Figure 4. In lines 249-250 we provided an explanation for the decrease of compound 3. We have now added two lines to explain also the decrease of compound 1 in Kuvan (lines 259-260). An arrow has also been added in Figure 4.
Figures captions need more explanation.
Answer: Following your suggestion, we have added more details to the captions of figures 1, 3, and 4. In this way, these figures are clear, independently from the main text.
References should be updated
Answer: We have replaced ref. 4 with a more recent one; in addition, we have added two new references (ref. n. 10 and 13)
In the analysis step, the author should mention or test the amount of the drugs after the sonication, and centrifuge processes.
Answer: To clarify this point, we have added the following sentence in lines 82-83 “Given the reported solubility of sapropterin in water (2.0 mg/mL) we could assume that it was fully solubilized in these conditions; however, in order to eliminate insoluble excipients, the solution was sonicated….”. The procedure of dissolution, sonication and centrifugation was repeated exactly in the same way for all the samples.
Changing from ascorbic acid-containing water to formic acid-containing water in MSMS should be discussed in more detail.
Answer: Actually, the sample was dissolved in water with 0.2 % ascorbic acid both for HPLC-UV (line 82) and LC-MSMS analysis (line 108). Moving from HPLC-UV to LC-MSMS we had to change the mobile phase buffer from phosphate to formate, since phosphate is not compatible with the Orbitrap MS instrument. However, in both cases the pH of the mobile phase was set around 3.0-3.5.
In such work, the solution of the 6R-(BH4) used as a source parameter. Please also check if it has impurities or containing other isomer or not.
Answer: Our standard sapropterin (6R-BH4) was obtained from Sigma Aldrich and it was highly pure and stable if conserved protected from light at –20 °C.
Reviewer 3 Report
The author developed an ion-exchange liquid chromatography method for the separation and analysis of Sapropterin and its 9 impurities, in which many pairs of isomers. And then the structures of these components were analyzed by high resolution mass spectrometry. The methods adequately described and the research design appropriate. Minor revisions should be made before acceptance of the manuscript.
1. In this study, the author mentioned that the reference substances was used to verify the impurities 1-9. The HPLC-UV chromatogram of these reference substances and the TIC of high resolution mass spectrometry shoud be provided.
2. Impurities 3 and 4, impurities 6 and 7 are two pairs of isomers. The m/z, MF and RDB between isomers are the same. How did the authors determine the peak order of these components on the chromatogram.
Reviewer 4 Report
In this study, a HPLC method was developed for the analysis of impurities in sapropterin tablets, and the impurities were identified by MS analysis. The manuscript may be more suitable for the journals specific in the pharmaceutical analysis such as Current Pharmaceutical Analysis.
1, The 1st and 2nd, as well as 3rd and 4th paragraphs of introduction section can be combined into one paragraph, respectively.
2, The abbreviations should be defined at their first used in the manuscript.
3, Table 1 and Figure 3 can be deleted, and describe the data in the text.
4, The unit for centrifugation can be convert to centrifugation force (×g), and please provide details of the centrifuge. Please also carefully check about the information for the materials and instruments that were used in this study.
Author Response
Reviewer 4
In this study, a HPLC method was developed for the analysis of impurities in sapropterin tablets, and the impurities were identified by MS analysis. The manuscript may be more suitable for the journals specific in the pharmaceutical analysis such as Current Pharmaceutical Analysis.
Answer: We respect the opinion of the reviewer. However, Pharmaceutics has already published studies on the HPLC-based evaluation of drugs stability (e.g. Pharmaceutics, 2017, 9, 11)
1, The 1st and 2nd, as well as 3rd and 4th paragraphs of introduction section can be combined into one paragraph, respectively.
Answer: The 1st and 2nd as well as 3rd and 4th paragraphs of Introduction have been combined
2, The abbreviations should be defined at their first used in the manuscript.
Answer: We have checked the entire manuscript and defined four additional abbreviations
3, Table 1 and Figure 3 can be deleted, and describe the data in the text.
Answer: Following the indication of the reviewer, Table 1 has been deleted and information contained have been incorporated into the text (lines 63-66). The following two Tables have been renumbered. On the other hand, we would keep Figure 3 since it visually depicts the overall results of our investigation.
4, The unit for centrifugation can be convert to centrifugation force (×g), and please provide details of the centrifuge. Please also carefully check about the information for the materials and instruments that were used in this study.
Answer: The Unit for centrifugation force has been provided as well as details on the centrifuge (lines 83 and 84). We have checked and improved all the materials information.
Round 2
Reviewer 2 Report
March 07, 2020.
Journal: Pharmaceutics-723825v2
Title: HPLC-Based Analysis of Impurities in Sapropterin Branded and Generic Tablets
The authors are presenting an analytical chromatographic method to separate and quantify impurities found in sapropterin-containing drugs (Diterin and Kuvan). HPLC and MS analysis are the tools used. The manuscript is recommended for publication after minor revision.
Comments to Authors:
- Could you add the originality of the work
- Please add the importance of this study in economic issues
- The authors could illustrate the effect of impurities on sapropterin-containing drugs` efficacy.
- Could you add similar work in the discussion
Author Response
- Could you add the originality of the work Answer: A sentence was alredy present in lines 223-225. A new sentence about the originality of the work has been added at lines 274-275
- Please add the importance of this study in economic issues Answer: Since it reports the comparison between two commercial products, our study has an evident impact in economic issues. However, our study has purely scientific aims and we would like to keep economic discussions out.
- The authors could illustrate the effect of impurities on sapropterin-containing drugs` efficacy. Answer: This was already discussed in lines from 277 to 283
- Could you add similar work in the discussion Answer: As far as we know references 11, 12, 13, already cited, are the only available similar works.
Reviewer 4 Report
The manuscript has been improved, and the comments of the reviewers have been well addressed. Figure 3 can be deleted, as the data was provided in Table 2.
Author Response
The manuscript has been improved, and the comments of the reviewers have been well addressed. Figure 3 can be deleted, as the data was provided in Table 2.
Answer: The reviewer is right. However, since Figure 3 visually depicts the results of Table 2 and can be used for rapid promotion of the article, we would like to keep it. This is in agreement with the policy of the journal that, as Open Access online journal, does not put space limitations.